# HPV Molecular Genotyping as a Differential Diagnosis Tool in Cervical Cancer Metastasis

**DOI:** 10.3390/jpm13020177

**Published:** 2023-01-19

**Authors:** Rosa Montero-Macías, Pluvio J. Coronado, Nicolas Robillard, David Veyer, Vincent Villefranque, Marie-Aude Le Frére-Belda, Elisabeth Auberger, Pauline Bitolog, Ivana Stankovic, Laurent Bélec, Anne-Sophie Bats, Fabrice Lécuru, Hélène Péré

**Affiliations:** 1Obstetrics and Gynecology Department, Centre Hospitalier Simone Veil, 95602 Eaubonne, France; 2Complutense University of Madrid, 28223 Madrid, Spain; 3Women’s Health Institute José Botella Llusiá, Fundación de Investigación del Hospital Clínico San Carlos (IdISSC), Universidad Complutense, 28040 Madrid, Spain; 4Virology Laboratory, Georges Pompidou European Hospital, 75015 Paris, France; 5INSERM, Functional Genomics of Solid Tumors (FunGeST), Centre de Recherche des Cordeliers, Université de Paris and Sorbonne Université, 75013 Paris, France; 6Pathology Department, Georges Pompidou European Hospital, 75015 Paris, France; 7Pathology Department, Simone Veil Hospital, 95600 Eaubonne, France; 8Faculty of Medicine, Paris University, 75015 Paris, France; 9Gynecologic and Breast Oncologic Surgery Department, Georges Pompidou European Hospital, 75015 Paris, France; 10Breast, Gynecology and Reconstructive Surgery Unit, Curie Institute, 75005 Paris, France

**Keywords:** HPV, cervical cancer, metastasis, HPV commercial genotyping assays, differential diagnostic tool

## Abstract

Background: Differentiating metastatic cervical cancer from another primary tumor can be difficult in patients with a history of cervical cancer and a distant lesion. The use of routine HPV molecular detection and genotyping tests could help in these cases. The objective of this study was to identify if an easy-to-use HPV molecular genotyping assay would allow differentiating between HPV tumor metastasis and a new independent primary non-HPV-induced tumor. Materials and Methods: Between 2010 and 2020, we identified patients with a primary cervical carcinoma who also had another secondary lesion. This identification included a clinical and histologic differential diagnosis of metastatic cervical cancer versus a new primary cancer or metastatic cancer from another site. We used a routine multiplex real-time PCR (rt-PCR) Anyplex^TM^ II HPV28 (Seegene, Seoul, Republic of Korea) to detect the high-risk (HR)-HPV genome in the distant lesions in these patients. Results: Eight cases of cervical cancer with a new secondary lesion were identified. In seven, HR-HPV DNA was detected in the biopsy of the distant lesion, which confirmed the diagnosis of cervical cancer metastasis. In the remaining case, no HPV was detected in the secondary lung biopsy, confirming the diagnosis of new primary lung cancer. Conclusion: Our results pave the way for HPV molecular genotyping use in cases of newly diagnosed distant lesions in patients with a history of HPV cervical neoplasia by using a routine diagnosis process to complete the clinical and histologic differential diagnosis when confronted with ambiguous situations.

## 1. Introduction

Cervical cancer induced by high risk (HR)-human papilloma virus (HPV) infection is the fourth most frequent cancer in women worldwide with 569,000 new cases each year [1]. Even in early clinical stages, cervical cancer can be complicated by lymph nodes metastasis or a hepatic, pulmonary or abdominal distant lesion implying a worse prognosis [2]. In addition, women with cervical cancer are at increased risk for other HPV-related carcinomas [3]. Patients with primary HPV-associated cervical cancer may have an increased risk of developing secondary HPV-related malignancies given that HPV can cause precancerous lesions and invasive malignancy when it infects oro-genital mucosa and genital skin [4,5]. Thus, some cervical cancer patients can present a distant lesion that may signify either metastatic disease from the primary cervical cancer or a new primary tumor or metastasis of another form of cancer.

Lesions of unknown origin out of the genital tract could occur in cervical cancer patients. Differential diagnosis between these lesions in the pathological analysis can be difficult, even with the help of immunohistochemistry techniques (IHC). However, the right identification of their origin will be essential for patients’ clinical care, treatment and prognosis.

Since HPV presence is required to maintain the tumoral phenotype, [6] HPV DNA could be assimilated to tumoral DNA and represent an interesting marker of the presence of tumor cells. HPV DNA should be detectable in metastatic tissue from primary tumors induced by HPV. Therefore, it prompts us to use our routine multiplex HPV genotyping real-time PCR (rt-PCR) Anyplex^TM^ II HPV28 (Seegene, Seoul, Republic of Korea) to study the HR-HPV genome in new distant lesions in cervical cancer patients. The objective of this study was to identify if such an easy-to-use HPV molecular genotyping assay would allow differentiating HPV tumor metastasis from a new independent primary non-HPV-induced tumor to improve care management of patients.

## 2. Material and Methods

### 2.1. Patients and Data Collection

We retrospectively included women with a histologically confirmed primary cervical carcinoma treated with surgery and/or chemoradiotherapy who also had another distant lesion. Patients were treated in two French medical centers (George Pompidou European Hospital and the Simone Veil Hospital) between 2010 and 2020. The lesion was suspected based on clinical symptoms or the results from imaging methods, namely magnetic resonance imaging (MRI), positron emission tomography (PET) or computed tomography scan (CT). All new lesions were biopsied and assessed by specialized pathologists in gynecological tumors.

Clinical and histologic differential diagnoses for the distant lesion were performed in order to differentiate whether the lesion was a metastasis from either the known cervical cancer or caused by another primary/metastatic cancer. Classical HE (Hematoxylin Eosin Safran) and IHC analyses were performed. In most cases, the histological type and IHC were compatible with different origins (metastasis or other primary cancer), allowing for multiple diagnostic options.

The diagnosis of any secondary lesions occurred at the same time as that for primary cervical cancer (synchronous), or later on during follow-up. For any secondary lesion, we recorded the year of diagnosis, location of the lesion, histology type and the p16 expression status (Table 1).

### 2.2. HPV Status in Primary Tumors and Distant Lesions

Formalin-fixed, paraffin-embedded (FFPE) biopsies of distant lesions and the correspondent available tumor (only available for 4/8 patients) were sent to the ISO 15189-accredited virology laboratory of the Georges Pompidou European Hospital (Paris, France) for DNA extraction as described by Steinau et al. [7] prior to HPV detection and genotyping using the Anyplex™ II HPV28 genotyping test (Seoul, Republic of Korea). Sections of the FFPE biopsies were deparaffinized overnight at 56 °C with 360 μL of ATL buffer (Qiagen, Hilden, Germany) and then we added 40 μL of proteinase K. Afterwards, 200 μL of ATL buffer was added then incubated for 10 min at 70 °C. DNA was further extracted using a QiaAmp DNA Mini Kit (Qiagen) and eluted in 50 μL of PCR-grade water. HPV detection and genotyping was carried out in 5 µL of extracted DNA using the CE IVD-marked multiplex rtRT-PCR assay Anyplex™ II HPV28 (Seegene, Seoul, Republic of Korea) [8,9]. The Anyplex ™ II HPV28 detection test distinguishes 28 HPV genotypes by amplifying 100–200 bp fragments of the L1 gene including 13 high-risk types (HR-HPV -16, -18, -31, -33, -35, -39, -45, -51, -52, -56, -58, -59 and -68), eight low-risk types (LR) (LR-HPV -6, -11, -40, -42, -43, -44, -54 and -70), seven genotypes reported as possibly carcinogenic (HPV-26, -53, -61, -66, -69, -73 and -82) as well as the human gene β-globin in two different reactions [9]. The DNA amplification and the genotyping process were carried out in two reactions performed on the CFX96^TM^ real-time PCR instrument (Bio-Rad, Marnes-la-Coquette, France) [8]. Melting curves were obtained at 30, 40 and 50 cycles. Data recording and interpretation were automated using Seegene Viewer software version 2.0 (Seegene) in accordance with the manufacturer’s instructions. Raw data of the results were checked by a virologist. This molecular HPV genotyping assay has been found to be suitable for HPV detection and genotyping in cervical secretions [8,9,10,11]. Based on Seegene’s proprietary DPO^TM^ and MuDT^TM^ technologies [12], this assay was conceived to avoid mismatch priming and to quantify each target in a single fluorescence channel, respectively. In addition, the Anyplex^TM^ II HPV28 assay was shown to be suitable for an extended genotyping approach with a high sensitivity in FFPE specimens [11,13].

## 3. Results

### 3.1. Clinical Characteristics and Pathological Findings of Patients

Under our inclusion criteria, we identified eight women treated in two French comprehensive cancer centers between 2010 and 2020 (Table 1).

In these eight cases, the secondary lesion was difficult to identify as a cervical tumor metastasis or second primary tumor.

The clinical and histological features of these women are described in Table 1. The mean age of the patients was 53.9 years (range 40–79). The most common histological type of tumor was squamous carcinoma (six cases). Two cases were adenocarcinoma. All cases were diagnosed in advanced stages (IIb or IVa 2018 FIGO stage).

Two cases were synchronous with primary cervical cancer. The other cases were diagnosed later (the time of recurrence/metastasis was between one and three years). Moreover, except for the synchronous cases that were treated with surgery, all other patients received chemoradiotherapy as initial treatment after the tumors were biopsied. The most common site of the second tumor was the lung (four cases).

### 3.2. HPV Detection and Genotyping of Distant Lesions FFPE Biopsies

For the results interpretation, the quality of extracted DNA was confirmed by the systematic positivity of the internal control β-globin for all samples. Therefore, in the distant lesion, results from RT-PCR showed HPV-16 DNA in six cases and HPV-18 DNA in one case, thus completing the histological results and confirming the cervical origin of the new lesion (Table 1). Lastly, a case with an HPV-negative pulmonary lesion completed our series, confirming that this new primary independent cancerous lesion was not related to the primary HPV cervical cancer in this patient. The HPV genotype was confirmed via biopsies of primary tumors in some of the cases (see #E, #F, #G and #H in Table 1).

## 4. Discussion

In this study, eight documented cases of women with cervical cancer showed that the use of routine HPV molecular detection and genotyping allowed the differentiation between metastatic cervical cancer disease and a new primary tumor.

We found HPV in all distant lesions except one case. This case, with negative HPV in the lesion, was considered as primary lung cancer and was treated with Navelbine with a fractionated dose of cisplatin (CDDP), unlike the cases diagnosed as metastatic cervical cancer, which were treated with carboplatin and taxol.

The prognosis of patients with metastatic cervical cancer is poor (median survival time between 8 and 13 months). There are two types of metastases in cervical cancer patients: hematogenous and lymphatic. Patients with hematogenous metastasis have a higher risk of mortality than those with lymphatic metastasis. In addition, the management of metastasis may be different depending on its origin and location. Surgery, radiotherapy, chemotherapy or combinations of several treatments are options [2].

In patients with HPV-related cancer and a distant lesion, the histologic differentiation of metastatic cervical cancer versus another primary tumor or metastasis from another cancer can be difficult, especially if the distant lesion shows similar histologic findings as the cervical cancer [14]. However, this differential diagnosis is essential in view of its clinical, therapeutic and prognostic implications. In addition, patients with HPV-related cancer are at higher risk of having other carcinomas since smoking is a common risk factor. In our study, four of the distant lesions were pulmonary, which raised diagnostic doubts as to whether they were primary lung cancer or metastasis of cervical cancer. In both cases, smoking could have been a risk factor and therefore a confounding factor in the differential diagnosis.

Overexpression of the cyclin-dependent kinase inhibitor gene p16^(INK4a)^ is a well-established surrogate marker in HPV-related malignancies [15]. p16 expression detection could help in differential diagnosis but is not a specific marker. Data published in 2016 have shown that primary ovarian cancer and borderline ovarian lesions can also express p16 with diffuse, moderate-to-strong p16 immunoreactivity [16]. Additionally, p16 expression has also been observed in different types of pulmonary carcinomas [17]. This made the diagnosis of our patients who had lung and ovarian lesions difficult since p16 positivity did not allow us to make a clear diagnosis of metastatic cervical cancer.

Techniques such as in situ reverse transcriptase polymerase chain reaction (RT-PCR) for human papillomavirus RNA and in situ hybridization have allowed the differentiation of metastatic cervical carcinoma from either a new primary tumor or metastasis from another primary tumor [14]. Although in situ RT-PCR allows direct correlation of the viral signal with the histologic features in the tissue, RNA extraction from FFPE biopsies remains difficult and inefficient due to degradation or modification of the RNA. This renders the RT-PCR process on such samples delicate and relatively laborious [18]. On the other hand, in situ hybridization is known to lack sensitivity compared to molecular techniques [14]. Lastly, both of these techniques do not precisely identify the detected HPV genotype.

Arfi et al. recently used the identification of the same HPV integration site in the paired DNA of endo-cervical and ovarian tumors as a proof of cervical metastasis in the ovaries [19]. Nevertheless, this technique remains much more complex and expensive compared to easy-to-use commercial HPV genotyping kits. Moreover, as HPV infection and associated tumoral processes only occur in mucosal tissue, the confirmation of correspondent tumoral HPV presence by simple, specific and sensitive molecular techniques in non-mucosal tissue appears to be sufficient to confirm the metastatic status of a secondary diagnosed lesion after onset of primary mucosal HPV cancer.

Conversely, in the literature, we did not find any cases of HPV detection in distant lesions that were unrelated to cervical cancer, so the possibility of a false positive from this technique seems low.

In our cases, we only identified HPV 16 or 18 in tumors and secondary lesions; however, other studies have detected other types, most frequently HPV 45 [19].

As our study was a retrospective analysis, the sample size presented is low and represents a clear limitation of our study. Indeed, we only succeeded in achieving HPV genotyping in half of the initial cervical tumors, which allowed us to match the HPV genotype from both the initial and the distant lesion, due, in some cases, to the poor preservation of DNA in old FFPE samples. It also highlighted the interest of systematically performing HPV molecular genotyping of cervical cancer to optimize post-therapeutic medical care and monitoring of such patients. In view of our results, we suggest that HPV molecular genotyping of newly diagnosed distant lesions in patients with a history of cervical neoplasia could be helpful to improve the diagnosis and treatment of such new lesions in patients with doubtful distant tumors.

## Figures and Tables

**Table 1 jpm-13-00177-t001:** Clinical and pathologic findings in patient cases of cervical cancer and possible metastatic disease.

Case	Age (Year)	Cervical Cancer Type	Multiplex rt-PCR**	New Lesions			
				Location	Histology	p16^(INK4a)^ Expression *	Multiplex rt-PCR **
#A	79	Squamous cellcarcinoma [stage IIB] ***in 2015	NA	Mediastinal adenopathies (2018)	Squamous cell carcinoma	+	HPV-16
#B	49	Adenocarcinoma [stage IIB] *** in 2017	NA	Lung, liver, brain, retroperitoneum (2018)	Undifferenciated carcinoma	+	HPV-16
#C	53	Adenocarcinoma [stage IVA] *** in 2018	NA	Ovary (2018,synchronous)	Endometrioid Adenocarcinoma	ND	HPV-18
#D	42	Squamous cell carcinoma [stage IIB] *** in 2019	NA	Ovary (2019,synchronous)	Squamous cell carcinoma	+	HPV-18
#E	47	Squamous cell carcinoma [stage IIB] *** in 2016	HPV-16	Lung (2019)	Squamous cell carcinoma	ND	HPV-16
#F	40	Squamous cell carcinoma [stage IIB] *** in 2017	HPV-16	Lung (2018)	Squamous cell carcinoma	+	HPV-16
#G	54	Squamous cell carcinoma [stage IIB] *** in 2010	HPV-16	Kidney (2012)	Squamous cell carcinoma	ND	HPV-16
#H	67	Squamous cell carcinoma [stage IIB] *** in 2016	HPV-16	Lung (2016)	Adenocarcinoma	ND	None

* p16^(INK4a)^ expression was assessed by immunohistochemistry using rabbit monoclonal anti-CDKN2A/p16^(INK4a)^ antibody; ** Sections of formalin-fixed, paraffin-embedded (FFPE) biopsies were deparaffinized overnight at 56 °C with 40 μL of proteinase K (Qiagen, Hilden, Germany) and 360 μL of ATL buffer (Qiagen), as described by Veyer, 2019. Afterwards, 200 μL of ATL buffer was added and incubated for 10 min at 70 °C. DNA was further extracted using a QiaAmp DNA Mini Kit (Qiagen) and eluted in 50 μL of PCR-grade water. HPV detection and genotyping was carried out in 5 µL of extracted DNA using the CE IVD-marked multiplex rtRT-PCR assay Anyplex™ II HPV28 (Seegene, Seoul, Republic of Korea) as described by Estrade 2014 and Lillsunde Larsson, 2015. The Anyplex ™ II HPV28 detection test distinguishes 28 genotypes of HPV, by amplifying 100–200 bp fragments of the L1 gene including 13 high-risk types (HR-HPV -16, -18, -31, -33, -35, -39, -45, -51, -52, -56, -58, -59 and -68), eight low-risk types (LR) (LR-HPV -6, -11, -40, -42, -43, -44, -54 and -70), seven genotypes reported as possibly carcinogenic (HPV-26, -53, -61, -66, -69, -73 and -82) as well as the human gene β-globin in two different reactions (Lillsunde Larsson 2015). The DNA amplification and the genotyping process were carried out in two reactions performed on the CFX96™ real-time PCR instrument (Bio-Rad, Marnes-la-Coquette, France) (Estrade 2014). Melting curves were obtained at 30, 40 and 50 cycles. Data recording and interpretation were automated using Seegene Viewer software version 2.0 (Seegene) in accordance with the manufacturer’s instructions. Raw data of the results were checked by a virologist. The virology laboratory was accredited in 2013 by the Comité Français d’Accréditation (COFRAC) according to ISO 15189 norms for the biological markers “HPV detection” and “HPV genotyping”. *** According to the International Federation of Gynecology and Obstetrics (FIGO) classification. Age is related to the distant lesion diagnosis. All patients were initially treated with radiochemotherapy. Distant lesions were treated with chemotherapy: carboplatin and taxol in all the cases and adding Vinorelbine in case H (primary lung cancer). ND, Not done; rt-PCR, real-time polymerase chain reaction; +, positive; NA, not available.

## Data Availability

Data is unavailable due to privacy or ethical restrictions.

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
