# Peer review of "HPV Molecular Genotyping as a Differential Diagnosis Tool in Cervical Cancer Metastasis"

_jpm, 2023, doi:10.3390/jpm13020177_

Round 1

Reviewer 1 Report (New Reviewer)

This study evaluated the effectiveness of HPV test for the differential diagnosis of metastatic cervical cancer from another primary cancer with cervix metastasis.

Since there were no similar studies, this study has its own originality. Study is well-designed, and manuscript is well-written. Conclusion is very clear and discussion is well-written and easy to read.

Author Response

We really want to thank again the reviewers for their time to examine this new version and their very pertinent commentaries and suggestions.

Reviewer 2 Report (New Reviewer)

Sample size presented may be a key drawback, but this limitation may be accepted with a clearer justification by the authors. Results should give details about the quality of DNA obtained from FFPE sections which was used for HPV detection. This is mainly to explain to interested readers why results of RT-PCR from primary tumors are not available for four patients since the materials and methods section gives the notion that primary and distant lesions were obtained from all eight patients evaluated. 

Author Response

Reviewer 2:

  1. Sample size presented may be a key drawback, but this limitation may be accepted with a clearer justification by the authors.

We are totally agree with this remark and add a sentence to justify the sample size in the discussion (Lines 297,298)

  1. Results should give details about the quality of DNA obtained from FFPE sections which was used for HPV detection. This is mainly to explain to interested readers why results of RT-PCR from primary tumors are not available for four patients since the materials and methods section gives the notion that primary and distant lesions were obtained from all eight patients evaluated. 

Indeed, The Reviewer 2 is absolutely right and the quality of the DNA can be poor, especially in old samples, which partly explains the lack of HPV results in some primary tumors. Ti explain that more clearly for interested readers, we add some explanations in the Material and Methods section (lines 116-117), in theResults section (lines 214-216) and in the Discussion section (lines 301,302).

This manuscript is a resubmission of an earlier submission. The following is a list of the peer review reports and author responses from that submission.

Round 1

Reviewer 1 Report

This was a case report that containing the doubtful statement. The author presented the eight cases of advanced stage cervical cancer with other site lesion. Only one case (#H) that the author claimed that the lesion at the cervix and lung lesion were different due to the negative result of P16 staining and HPV test the distant site. It was inclusive to claim that this was the synchronous tumor of cervix and lung. Most synchronous tumor was usually early stage of both sites and having good prognosis. The only one case in this study cannot differentiated between synchronous cases or false negative test of HPV at the lung lesion.